

Factors controlling spatiotemporal variability of soil carbon accumulation and stock estimates in a tidal
salt marsh
Author Information:
[1,5] Sean Fettrow ORCID: 0000-0003-1191-4484
[2] Andrew Wozniak ORCID: 0000-0002-7079-3144
[3,4] Holly A. Michael ORCID: 0000-0003-1107-7698
[1,3] Angelia L. Seyfferth ORCID: 0000-0003-3589-6815
[1]Department of Plant and Soil Sciences, University of Delaware, Newark DE, USA
[2]School of Marine Science and Policy, University of Delaware, Lewes DE, USA
[3]Department of Earth Sciences, University of Delaware, Newark DE, USA
[4]Department of Civil and Environmental Engineering, University of Delaware, Newark, DE, USA
[5]Oak Ridge National Laboratory, Environmental Sciences Division
* Corresponding Author: Angelia Seyfferth, angelias@udel.edu





**Abstract**
Tidal salt marshes are important contributors to soil carbon (C) stocks despite their relatively small land
surface area. Although it is well understood that salt marshes have soil C burial rates orders of magnitude
greater than those of terrestrial ecosystems, there is a wide range in storage rates among spatially
distributed marshes. In addition, wide ranges in C storage rates also exist within a single marsh
ecosystem. Tidal marshes often contain multiple species of cordgrass due to variations in hydrology and
soil biogeochemistry caused by microtopography and distance from tidal creeks, creating distinct subsites.
Our overarching objective was to observe how soil C concentration changes across four plant
phenophases and across three subsites categorized by unique vegetation, hydrology, and biogeochemistry,
while also investigating dominant biogeochemical controls on soil C concentration. We hypothesized that
subsite biogeochemistry drives spatial heterogeneity in soil C concentration, and this causes variability in
soil C concentration at the marsh scale. In addition, we hypothesized that soil C concentration and
porewater biogeochemistry vary temporally across the four plant phenophases (i.e., senescence,
dormancy, green-up, maturity), causing further variation in marsh soil C that could lead to uncertainty in
soil C estimates. To test these hypotheses, we quantified soil C concentrations in 12 cm sections of soil
cores (0-48 cm depth) across time (i.e., phenophase) and space (i.e., subsite), alongside several porewater
biogeochemical variables including dissolved organic carbon (DOC), EEMs/ UV-VIS, redox potential,
pH, salinity, reduced iron ($Fe^{2+}$), reduced sulfur ($S^{2-}$), and total porewater element (Fe, Ca) concentrations
in three distinct subsites. Soil C concentration varied significantly ($p<0.05$) among the three subsites and
was significantly greater during plant dormancy. Soil S, porewater sulfide, redox potential, and depth
predicted 44% of the variability in soil C concentration. Our results show that soil C varied spatially
across a marsh ecosystem up to 63% and across plant phenophase by 26%, causing variability in soil C
storage rates and stocks depending on where and when samples are taken. This shows that hydrology,
biogeochemistry, and ecological function are major controls on saltmarsh C content. It is, therefore,
critical to consider spatial and temporal heterogeneity in soil C concentration when conducting blue C
assessments to account for soil carbon variability and uncertainty in C stock estimates.




## 1 Introduction

Coastal blue carbon (C) cycled in tidal salt marshes is critically important for global soil C
sequestration despite the small relative land area (Mcowen et al. 2017). High primary productivity
coupled with high sedimentation rates and slowed organic C decomposition due to flooded anoxic soils
allow salt marshes to rapidly accrete and preserve soil C (Arias-Ortiz et al. 2018). Soils in such
ecosystems retain approximately 15% of their yearly primary productivity in soils compared to just 1%
for tropical rainforests (Duarte 2017). Restoring, protecting, and artificially creating salt marshes can
facilitate removal of $CO_2$ from the atmosphere and storage in soils on timescales conducive to climate
change mitigation goals. These ecosystems should therefore be included in climate mitigation policy
(Ewers Lewis et al. 2019; Serrano et al. 2019). However, a wide range of global salt marsh soil C
sequestration rates of ~ 1 to >1100 g C m$^{-2}$ year$^{-1}$ has been reported (Wang et al. 2021). The inclusion of
salt marshes in improved climate mitigation policy is, in part, contingent upon improving our
understanding of the environmental variables causing wide ranges in marsh soil C concentration and thus
soil sequestration rates (Saintilan et al. 2013; Macreadie et al. 2019). Understanding key controls on salt
marsh soil C variability will also decrease uncertainty in Earth System Models and inform new policy
aimed at protecting these valuable ecosystems.
Soil C concentrations in salt marsh ecosystems vary spatially across the globe. Part of this
variation is explained by regional environmental controls such as average annual air temperature (Chmura
et al. 2003), geomorphic setting (van Ardenne et al. 2018), salinity gradients, inundation frequency (van
de Broek et al. 2016; Baustian et al. 2017; Luo et al. 2019), rainfall patterns (Sanders et al. 2016;
Negandhi et al. 2019), soil pH, soil moisture, and the dominant plant species and soils (Bai et al. 2016;
Ford et al. 2019). Soil C accumulation rates also vary based on the age of the marsh and tend to be highest
in newly expanding marsh edges (Miller et al. 2022). Other logistical factors contributing to variability
and heterogeneity in salt marsh blue C estimates include the type of corer used (Smeaton et al. 2020) and



the depth of soil that is integrated into storage rates (Bai et al. 2016; Van De Broek et al. 2016; Mueller et
al. 2019). While understanding global and regional controls on soil C is important for reducing
uncertainty in C estimates, understanding site-level factors is also critical because ecosystem-level
variability can be just as high as regional- to global-level variability (Ewers Lewis et al. 2018).
Belowground biogeochemical heterogeneity is often noticeable in the aboveground vegetation due to
striking zonation of marsh grass species across the marsh platform. This is often attributable to small
spatial-scale changes in hydrologic patterns (Guimond et al. 2020b, a) based on proximity to the tidal
channel that drives unique subsite biogeochemistry (Seyfferth et al. 2020) which subsequently determines
the type of vegetation that can survive within a given tidal zone (Davy et al. 2011). While tidal zonation
alters vegetation and belowground biogeochemistry, it remains unclear if soil C concentrations are
directly or indirectly altered by these dynamics.

Primary production rates may partially control soil C concentration and may vary among

vegetative zones. For example, the short form of *Spartina alterniflora* has a lower primary production rate
than the tall form (Roman and Daiber 1984) and *Phragmites australis* has above and below ground
production rates two times that of the shorter *Spartina patens* (Windham 2001). Belowground
productivity includes root exudates (Luo et al. 2018) in the form of dissolved organic carbon (DOC),
which could influence soil C concentration because belowground productivity often exceeds above
ground productivity in these ecosystems (Frasco and Good 1982). Even though DOC exudates are
considered to be labile (Yousefi Lalimi et al. 2018), they may contribute to soil C accumulation over time
due to microbial transformation (Valle et al. 2018) and association with soil minerals such as Fe oxides
(Chen et al. 2014; Chen and Sparks 2015; Sowers et al. 2018a, b, 2019). The characterization of DOC
quantified by optical properties of chromophoric dissolved organic carbon (CDOM) can also affect
degradability (Clark et al. 2014) and may differ across the marsh platform.

Subsites can have unique biogeochemical signatures based on soil redox conditions and

inundation extent and frequency. For example, high marsh areas and areas near tidal channels have soils
which are at least periodically oxic to sub-oxic and are dominated by iron (III) reduction, whereas low



marsh areas have continuously inundated soils and are dominated by sulfate ($SO_4^{2-}$) reduction (Seyfferth
et al. 2020). While these biogeochemical characteristics can directly influence vegetation (Moffett and
Gorelick 2016) and thus indirectly influence soil C concentrations, they may also directly affect soil C
through the interactions of soil C with soil minerals. Fe oxides have an intimate role in the C cycle and C
stabilization in soils experiencing dynamic redox fluctuation (Sodano et al. 2017), as previous work has
shown that 99% of the dissolved Fe in the ocean is complexed with organic ligands (Whitby et al. 2020)
and ~21% of all organic C in marine sediments is bound to reactive Fe species (Lalonde et al. 2012). Fe
oxides may play an important role in C stabilization in soils experiencing dynamic redox fluctuation. Fe
oxides can protect DOC against microbial degradation through physiochemical protection (Blair and Aller
2012; Chen and Sparks 2015; Sodano et al. 2017; Sowers et al. 2018a; Dorau et al. 2019; Wordofa et al.
2019), but these organo-mineral assemblages can be dissociated under reducing conditions (Riedel et al.
2013; Wordofa et al. 2019; Lacroix et al. 2022; Fettrow et al. 2023a). Therefore, examining the spatial
variability in soil biogeochemistry and relating those variables to soil C concentration may elucidate
important mechanisms that cause the wide range in salt marsh soil C concentrations.

While it is critical to assess spatial heterogeneity in soil C concentration, it is also important to

assess temporal variability. The temporal assessment of soil C in salt marshes often considers long-term
trends of historic C burial rates (Cusack et al. 2018; McTigue et al. 2019; Breithaupt et al. 2020; Cuellar-
Martinez et al. 2020), but variability of salt marsh soil C concentrations may also occur on shorter time
scales such as across a single year. Several studies suggest salt marsh soil C does not significantly change
across seasons throughout the year (Yu et al. 2014; Zhao et al. 2016), even though major changes in soil
biogeochemical variables occur on this timescale (Koretsky et al. 2005; Negrin et al. 2011; Seyfferth et al.
2020; Trifunovic et al. 2020; Zhu et al. 2021). While soil C concentration may be stable across seasons, it
is unclear if soil C concentration changes based on site-specific plant phenology. The phenophase of a
marsh is associated with the greenness index of vegetation (Trifunovic et al. 2020) and is strongly
associated with C dynamics in wetland systems (Desai 2010; Kang et al. 2016). Soil C concentration



should be measured across plant phenophase to determine if temporal changes in phenology alter soil C
concentration and cause another source of variability in ecosystem-scale C estimates.

To address these knowledge gaps, we conducted a year-long study of a temperate tidal salt marsh

to assess how soil C concentration and porewater biogeochemistry change in space (subsite) and time
(phenophase). Our overarching research objectives were to understand how soil C concentration and soil
biogeochemistry change across spatial and temporal scales, and to investigate key biogeochemical
mechanisms influencing soil C concentration at the ecosystem level. We hypothesized that subsites would
contain significantly different concentrations of soil C due to differences in soil biogeochemistry across
the marsh platform. We further hypothesized that soil C concentration and associated biogeochemistry
would significantly differ across plant phenophase. Our results improve understanding of mechanistic
controls on salt marsh soil C with implications for characterizing and reducing uncertainty in C
sequestration estimates, while also adding to the body of literature that shows tidal salt marshes are
critical reservoirs of sequestered C.
**2.0 Methods and Materials**
**2.1 Field Site**

This study was conducted at the St. Jones National Estuarine Research Reserve located in Dover,

Delaware (Figure 1). The ecosystem is classified as a temperate mesohaline tidal salt marsh with a tidal
creek salinity ranging from 5 to 18 ppt (Capooci et al. 2019). Three separate subsites were previously
identified at this site, each with a different vegetation type and hydrology (Guimond et al. 2020a; Seyfferth
et al. 2020). The subsite nearest the channel is primarily colonized by the tall form of *Spartina alterniflora*
and has semidiurnal tidal oscillation. This subsite is hereafter referred to as Tall Spartina (TS). Farther from
the tidal channel, the elevation is slightly higher due to a natural levee and flooding of the upper 25 cm of
soil occurs only during spring tides; this location has the larger cordgrass *S. cynosuroides* and is hereafter
referred to as Tall Cordgrass (TC). The third subsite is farthest from the tidal channel, lowest in elevation,
and is primarily colonized by the short form of *S. alterniflora* due to near continuous inundation; this subsite
is hereafter referred to as Short Spartina (SS). These subsites have distinct hydro-biogeochemistry and





vegetation that varies across small spatial scales and thus provides an ideal setting to understand site-level
variability in soil C concentration, porewater biogeochemistry and their relationships.

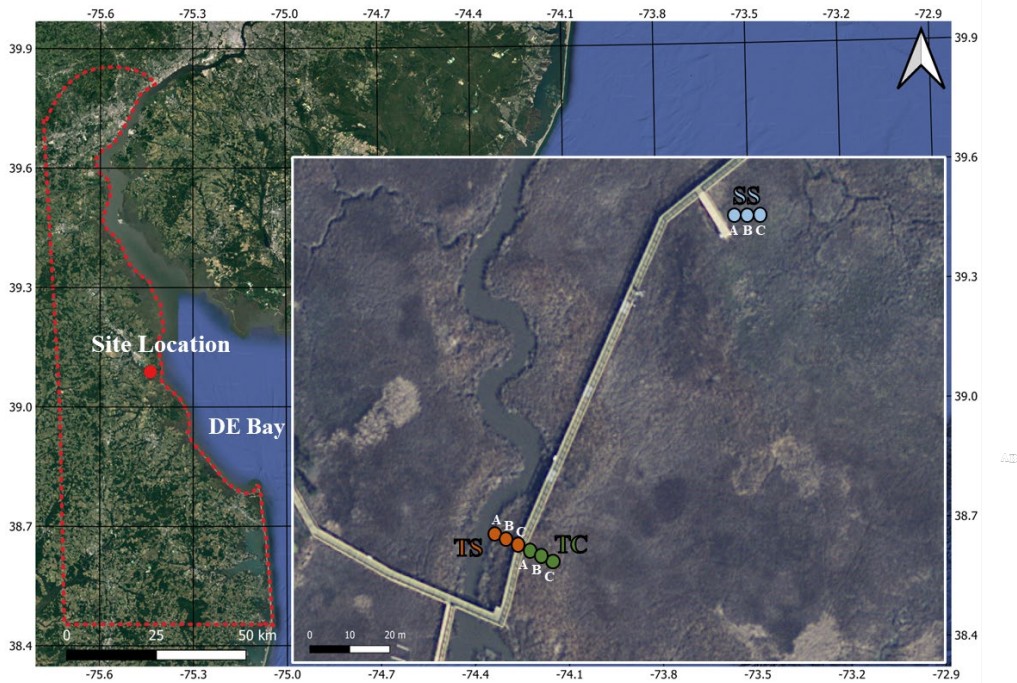


**Figure 1.** Map of the field site located at the St Jones Reserve near Dover, DE. Three unique subsites (TS,
TC and SS) have been characterized based on previous studies at this field site showing subsite specific
hydrology, vegetation, and biogeochemistry based on distance from the tidal creek (Guimond et al. 2020a;
Seyfferth et al. 2020). The coring locations were sampled in triplicate (Core A, B and C), with core A
starting closest to the creek and each subsequent core in each subsite being ~30cm from one another. The
base layer for the map was obtained from public base layers in QGIS ( © Google Maps).

**2.2 Soil Sampling and Analysis**

Soil cores were obtained from each of the three subsites (TS, TC, SS) in triplicate during each
sampling event. Replicates were taken approximately 30cm from one another and are labeled cores A, B,
and C based on distance to the tidal channel with A being closest to the channel and C the farthest (Figure
1). Sampling events occurred at four separate times of the year to coincide with each of the phenophases
(i.e., senescence on 10/3/2019, dormancy on 12/3/2019, green-up on 4/29/2020, maturity on 8/13/2020),
which were previously determined using the Greenness Index (Trifunovic et al. 2020). Soil cores (6 cm x



48 cm) were extracted using a gouge auger that has been shown to be an effective coring technique for
reducing compaction in soft marsh soils (Smeaton et al. 2020). Soil cores were quickly sectioned in the
field into 12 cm increments (0-12cm, 12-24cm, 24-36cm and 36-48cm relative to the soil surface) and
preserved under anoxic conditions following previous methods (Seyfferth et al. 2020). For reference, the
rooting zone of *Spartina* grasses is between 8-20cm (Muench and Elsey-Quirk 2019), so the upper two
sections likely include C from fresh root exudates. The 12cm increments were chosen because many soil
C stock papers use increments between 10-15 cm and there tends to be little variation across the ~10 cm
increment in a variety of wetland soils (Baustian et al. 2017). Briefly, the soil sections were placed into
250 ml HDPE bottles which were left uncapped in gas-impermeable bags that contained oxygen scrubbers
(AneroPack-Anero, Mitsubishi), and the bags were vacuum-sealed in the field. The soil samples were
placed on ice during transport back to the lab. Once back in the lab, the soil sections in the gas-
impermeable bags were immediately placed inside an anoxic glove bag containing ~5% hydrogen and
~95% nitrogen. A subsample of soil was dried, ground, sieved (2mm), and powdered for analysis of total
C and S (Vario EL Cube, Elementar). Soil C and S are reported as % C (= 100% * g C/g soil dry wt.) and
% S (= 100% * g S/g soil dry wt.). The remaining field-moist soil was left inside the HDPE vial, capped
inside the glove bag, and centrifuged for extraction of porewater using methods in the following section.
**2.3 Porewater Extraction and Analysis**
Porewater was extracted from each 12-cm soil section by centrifugation for 2 minutes under an
anoxic atmosphere at 2,500 rpm. A portion of the porewater was filtered with 0.45μm PTFE syringe
filters while the rest was vacuum filtered using glass fiber filters (0.7μm). The 0.45μm PTFE filtered
porewater was immediately analyzed for $Fe^{2+}$ using the ferrozine colorimetric method (Stookey 1970),
$S^{2-}$ using the methylene blue method (Cline 1969), redox potential with a 220mV offset, pH, and
conductivity using calibrated probes (Orion Ross Ultra pH/ATC Triode, Orion 9179E Triode, Orion
DuraProbe Conductivity Cell), and the remaining sample was acidified to 2% $HNO_3$ for elemental
analysis using an ICP-OES. The porewater filtered with glass fiber (0.7μm) was acidified with HCl and
analyzed for DOC (Vario TOC Analayzer, Elementar). To characterize the DOC, unacidified DOC



samples from the plant maturity sampling event were analyzed via ultraviolet-visible (UV-VIS)/
excitation-emission matrix spectroscopy (EEMs) (Aqualog Spectrophotometer, Horiba). The Aqualog
was zeroed with double deionized water blanks, checked using the manufacturer's excitation check,
corrected for inner filter effects, applied first and second order Rayleigh masking and data were
normalized using the average Raman area (Gao et al. 2011; Clark et al. 2014). Measurements were taken
over the wavelengths of 200-730nm with 2nm steps. Fluorescence and absorbance peaks and indices were
calculated using previously established equations (Table S1).
**2.4 Statistical Analysis**

Statistical differences between subsites and phenophase were analyzed using repeated measures

analysis of variance (ANOVA) ($\alpha=0.05$), with a post-hoc Tukey-HSD analysis to determine differences
between individual subsites and phenophase. Correlations with depth were analyzed using linear
regression and only the significant ($p<0.05$) relationships are reported. Relationships among all measured
variables were assessed using principal components analysis. In addition, a stepwise regression model
was built to determine variables that significantly predict soil C concentration. All statistical analyses
were conducted in JMP (Version 16.2).
**3.0 Results**
**3.1 Soil Carbon and Sulfur**

To explore the spatiotemporal heterogeneity of soil carbon (C) and sulfur (S) at each subsite,

subsamples of each collected soil increment were combusted for soil C and S concentration.
Concentrations of soil C were highly variable among subsites, phenophase, depth, and replicate cores
(Figure 2), indicating several possible sources of variability in marsh soil C stock estimates. SS showed
the highest soil C concentrations, as illustrated by darker colors in the heat map, compared to both TS and
TC. Soil C was also higher at TS than TC, illustrated by relatively darker colors in the heat map. For all
subsites, soil C concentrations changed throughout the year with the highest values during plant
dormancy and the lowest during green-up. However, variability across individual replicates A, B, and C
and with depth complicated generalities across time and space. For example, at subsite SS from 24-36 cm



during senescence, core A is ~5% soil C while core C is ~10% soil C, a factor of 2 difference within
replicates. Large ranges among replicates were also observed during green-up at TS from 12-24 cm and
during maturity at TC from 36-48 cm. This exemplifies the heterogeneity inherent in soils, and a source
of variation in marsh soil C estimates.

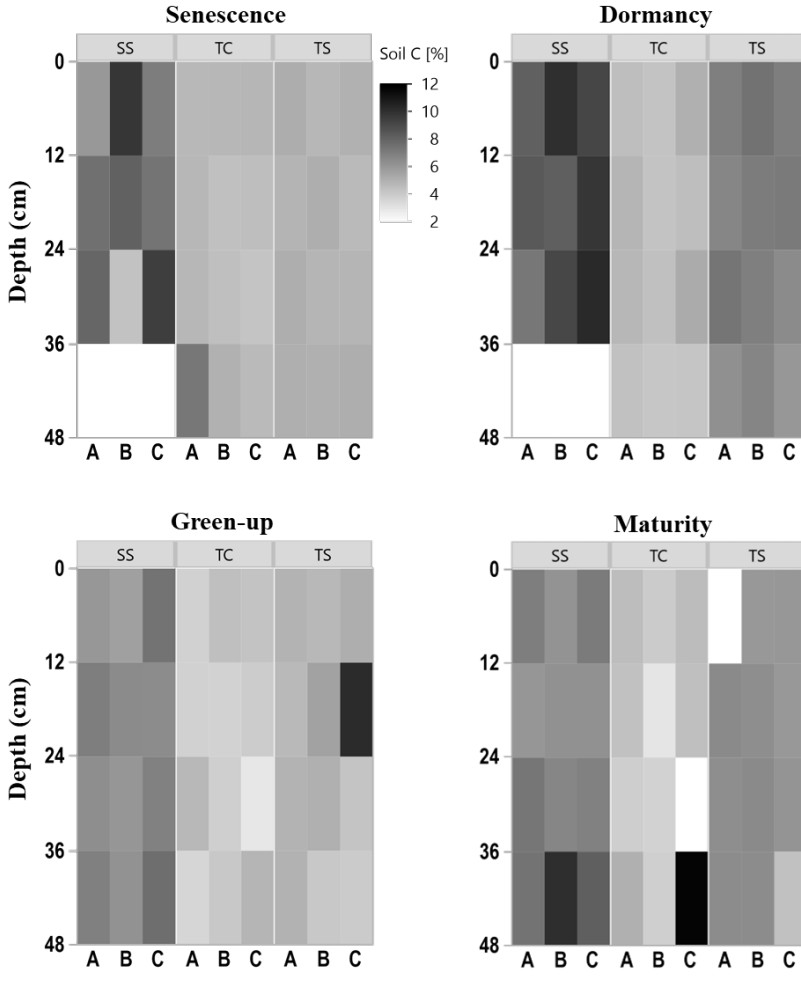


**Figure 2.** Heat maps of soil C concentration with depth at the three subsites (SS, TC, and TS), four
phenophases, and for each replicate core (A (closest to channel), B, and C (farthest from channel)). No
measurement was able to be obtained for some 12-cm sections as shown by white rectangles.
There was also variability in soil C concentration with depth (Figure 3). Subsite SS had the

highest mean soil C concentration at all four depths, as well as the largest range in values. TS had the





second highest mean soil C values at all four depths as well as the second largest range in values. TC had
the lowest mean soil C at all four depths as well as the smallest range in values at each depth. It is clear
from this graph that SS contains higher overall concentrations of soil C, followed by TS and then TC. Soil
C at TS during dormancy significantly decreased with depth ($R^2$=0.44, p=0.02) and soil C at SS during
maturity significantly increased with depth ($R^2$=0.41, p=0.02). No other correlations in soil C existed with
depth.

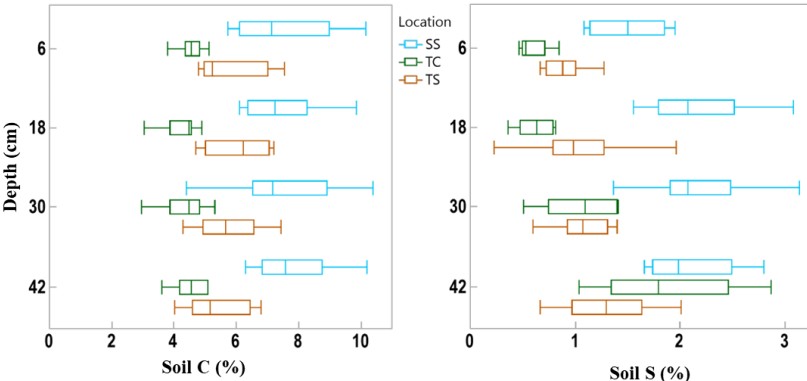


**Figure 3.** Box and whisker plot of soil C and S concentrations across the three subsites and separated by
the four sampling depths. This indicates the difference in soil C and S variability among subsites and with
depth. Whiskers indicate the minimum and maximum values, and the box indicates the upper and lower
quartiles. The line in the box indicates the median.
Soil S also varied across 12 cm sampling increment depths (Figure 3). SS had the highest mean
soil S concentration at each depth, and the range of values initially increased with depth. TS has a higher
mean concentration than TC at all depths except at the bottom core section. The range of soil S values
increased with depth at TC while the range was more consistent with depth at TS, except for the wide
range of values measured at the 18cm depth interval. Soil S at SS during maturity significantly increased
with depth ($R^2$=0.50, p=0.01), as did TC during dormancy ($R^2$=0.88, p<0.0001), green-up ($R^2$=0.51,
p=0.01), and senescence ($R^2$=0.42, p=0.02). No other correlations between soil S existed with depth.
**3.2 Porewater Data**
**3.2.1 Porewater DOC and Characterization**



Porewater DOC was highly variable across subsites, phenophase, depth, and replicate cores

(Figure 4). Note that the data in Figure 4 have been log transformed (natural log) due to large ranges in
values across the one-year sampling campaign. Unlike soil C, which was relatively consistent with depth,
DOC concentrations were highly variable with depth and even more so among replicate cores. Some of
the highest individual concentrations of DOC were detected nearest the surface and rooting zone, which
can extend to 20 cm below the surface (Muench and Elsey-Quirk 2019), but also at depth at SS during
senescence. DOC concentrations decreased with depth at SS during green-up ($R^2$=0.44, p=0.02) and
maturity ($R^2$=0.37, p=0.03) and increased with depth at TC during dormancy ($R^2$=0.76, p=0.0002). These
results indicate the highly variable nature of porewater DOC concentrations, possibly leading to
additional and complexity in marsh soil C estimates.



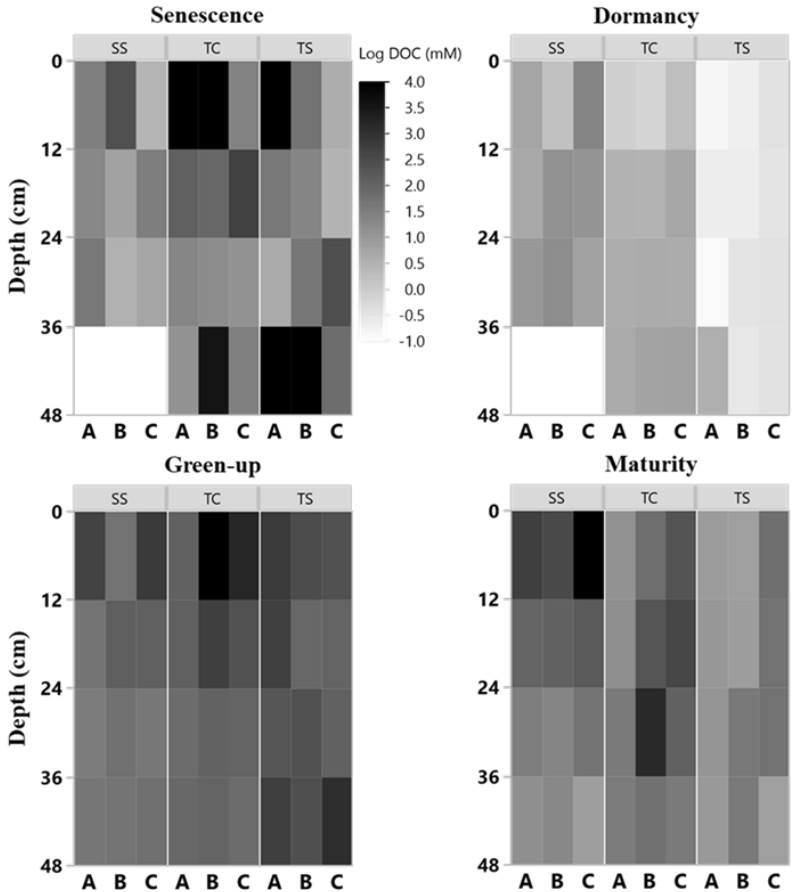


**Figure 4.** Heat maps of porewater DOC (natural log) concentration with depth at the three subsites (SS, TC, and TS), four phenophases, and for each replicate core (A (closest to channel), B, and C (farthest from channel)). No measurement was able to be obtained for some 12-cm sections as shown by white rectangles.

Porewater ultraviolet-visible (UV-VIS) and excitation emission matrices (EEMs) data were collected only from the maturity sampling event to further characterize DOC molecular properties (Figure 5). Optical properties (i.e., peaks, indices) from spectroscopic data were calculated and interpreted following previous studies cited in the supplemental table (Table S1). These data show significant trends with depth at SS. At SS, coble peak intensities T ($R^2$=0.55, p=0.01), B ($R^2$=0.49, p=0.01), A ($R^2$=0.57, p=0.004), M ($R^2$=0.55, p=0.01) and C ($R^2$=0.49, p=0.01) all significantly decreased with depth, as did the





fluorescence index (FI) ($R^2$=0.79, p=0.0001), the biological index (BIX) ($R^2$=0.50, p<0.01) and
absorbance at 254nm (Abs$_{254}$) ($R^2$=0.36, p=0.04), indicating decreases in CDOM with depth. To ensure
the coble peaks represented changes in CDOM properties and not DOC concentration, they were
normalized to DOC concentration and the relationships remained significant (p<0.05), except for the
Coble B peak ($R^2$=0.11, p=0.20). The $E_2$:$E_3$ ($R^2$=0.50, p=0.01) and SUVA$_{254}$ ($R^2$=0.53, p=0.007)
significantly increased with depth at SS, indicating a decrease in molecular weight and an increase in
aromaticity with depth. No significant trends with depth were present at TC or TS. Differences in DOC
molecular properties among subsites are apparent for many of the calculated indices and peaks.

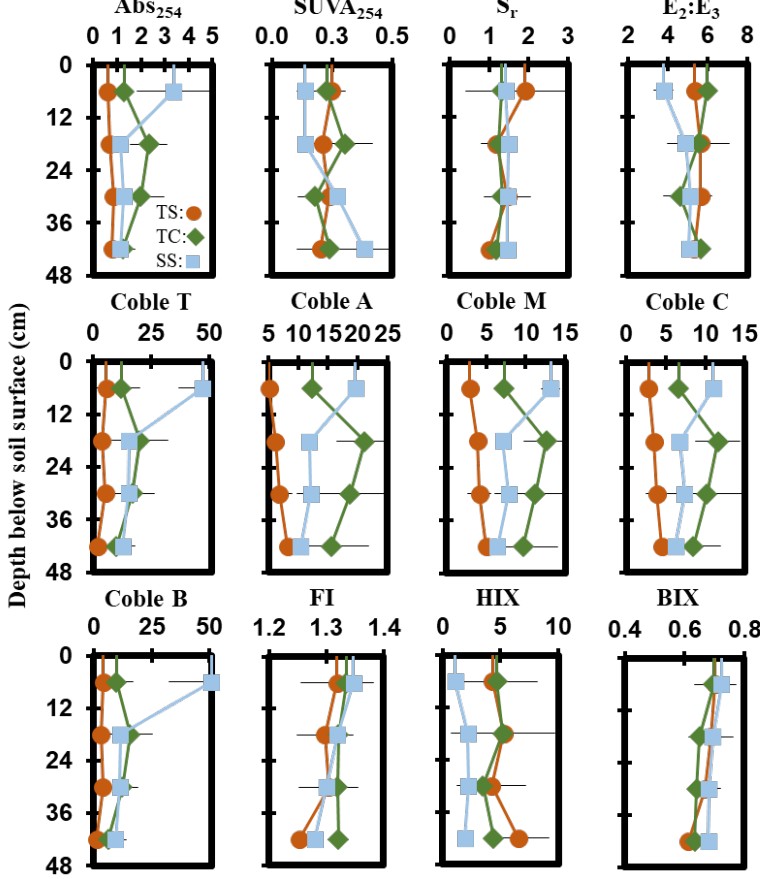


**Figure 5.** Depth profiles of porewater EEMs/ UV-VIS peaks and indices down to 48cm taken during the
maturity sampling event. Each point represents the mean between replicates (n=3) with error lines
indicating the standard deviation (± 1 SD).




### 3.2.2 Porewater Chemistry

Measured porewater biogeochemistry was variable across subsites, phenophase, and depth

(Figure 6). Porewater redox potentials showed minimal trends with depth, except for a significant
decrease with depth at SS during maturity ($R^2$=0.58, p=0.004), though redox showed variability between
replicates (Figure S2). The pH was relatively consistent with depth, except for a significant increase with
depth at TC during dormancy ($R^2$=0.42, p=0.02), and a significant decrease with depth at TS during
dormancy ($R^2$=0.56, p=0.005). Redox potential and pH formed a significant but weak negative correlation
($R^2$=0.12, p<0.0001) across the entire 1-year dataset.



**Figure 6.** Depth profiles of porewater chemistry variables down to 48cm for sampling events that occurred during plant (**a**) senescence, (**b**) dormancy, (**c**) green-up and (**d**) maturity. Each point represents the mean between replicates (n=3) with error lines indicating the standard deviation (± 1 SD).

Porewater $S^{2-}$ varied significantly with depth. $S^{2-}$ increased significantly with depth across the entire 1-year dataset ($R^2$=0.04, p=0.03). $S^{2-}$ increased significantly with depth at SS during green-up



($R^2$=0.51, p=0.01) and maturity ($R^2$=0.86, p<0.0001). TS $S^{2-}$ increased significantly during green-up
($R^2$=0.46, p=0.02) while TC $S^{2-}$ increased significantly during maturity ($R^2$=0.36, p=0.04). Porewater $Fe^{2+}$
trended negatively with $S^{2-}$ ($R^2$=0.06, p=0.004) and decreased with depth (p=0.01, $R^2$=0.05) across the
entire 1-year dataset. Significant decreases were observed at TS during green-up ($R^2$=0.68, p=0.001), and
at SS during maturity ($R^2$=0.41, p=0.02). Total Fe concentration followed similar depth trends to $Fe^{2+}$,
with a significant decrease with depth across the entire 1-year experiment ($R^2$=0.06, p=0.01). Total Fe
decreased with depth at TS during senescence ($R^2$=0.41, p=0.03) and green-up ($R^2$=0.58, p=0.004), and at
SS during maturity ($R^2$=0.57, p=0.01).

Porewater salinity formed varying relationships with depth. Salinity significantly decreased with

depth at TC during senescence ($R^2$=0.52, p=0.01), and at SS during maturity ($R^2$=0.62, p=0.002) while
salinity significantly increased with depth at TC during green-up ($R^2$=0.69, p=0.001) and at TS during
maturity ($R^2$=0.87, p<0.0001). Salinity and total Ca generally increased together (p>0.0001, $R^2$=0.42)
across the entire 1-year experiment. Total Ca increased significantly with depth at TC during green-up
($R^2$=0.86, p<0.0001) and at TS ($R^2$=0.80, p<0.0001) and TC ($R^2$=0.47, p=0.01) during maturity. SS total
Ca significantly decreased with depth during maturity ($R^2$=0.60, p=0.005).
**3.3 Analysis of Variance (ANOVA) Among Subsite and Phenophase**

ANOVAs were run on subsite and phenophase mean values that were obtained by averaging

samples from all depths across all four phenophases (for subsite comparisons) and all depths across all
three subsites (for phenophase comparisons). These results show significant spatial and temporal
variability in many of our measured variables. All three subsites contain significantly different average
concentrations of soil C, with SS having the highest average (7.5% C), followed by TS (5.8% C) and TC
(4.6% C). This indicates that on average, subsite SS contains ~29% more soil C than TS and 63% more
soil C than TC. In addition, plant dormancy contained significantly more soil C than plant green-up.
While soil S did not significantly vary across phenophase, soil S at SS was significantly higher in
concentration by a factor of two than both TS and TC.





**Table 1.** One-way ANOVA results for all assessed porewater biogeochemical variables. Mean values represent average values for each subsite for subsamples from all depths and phenophase. The mean is reported (± SD) along with a connecting letter report. Means with letters that do not connect are significantly ($p<0.05$) different.

| Variable | Tall Spartina (TS) | Tall Cordgrass (TC) | Short Spartina (SS) |
|---|---|---|---|
| Soil C (%) | 5.8±(1.2)$^B$ | 4.6±(1.3)$^C$ | 7.5±(1.4)$^A$ |
| Soil S (%) | 1.1±(0.5)$^B$ | 1.0±(0.6)$^B$ | 2.0±(0.7)$^A$ |
| DOC (mM) | 11.9±(27)$^A$ | 13.6±(27)$^A$ | 7±(9)$^A$ |
| Redox (mV) | 179±(176)$^{AB}$ | 211±(185)$^A$ | 93±(235)$^B$ |
| pH | 8.12±(0.8)$^A$ | 7.99±(0.7)$^A$ | 8.13±(0.6)$^A$ |
| $Fe^{2+}$ (mM) | 0.15±(0.1)$^A$ | 0.22±(0.3)$^A$ | 0.04±(0.1)$^B$ |
| Sulfide (mM) | 0.02±(0.01)$^B$ | 0.02±(0.01)$^B$ | 0.6±(0.6)$^A$ |
| Salinity (ppt) | 8.8±(3.1)$^B$ | 9.7±(3)$^{AB}$ | 11±(2)$^A$ |
| Total Fe (mM) | 0.21±(0.2)$^A$ | 0.26±(0.3)$^A$ | 0.08±(0.1)$^B$ |
| Total Ca (mM) | 4.7±(1.3)$^B$ | 5.4±(1.2)$^A$ | 5.8±(0.8)$^A$ |



**Table 2.** One-way ANOVA results for all assessed porewater biogeochemical variables. Mean values
represent average values for each phenophase for subsamples from all depths and subsites. The mean is
reported (± SD) along with a connecting letter report. Means with letters that do not connect are
significantly (p<0.05) different.

| Variable | Senescence | Dormancy | Green-up | Maturity |
|---|---|---|---|---|
| **Soil C (%)** | $5.7\pm(1.5)^{AB}$ | $6.7\pm(1.1)^{A}$ | $5.3\pm(1.5)^{B}$ | $6.1\pm(1.8)^{AB}$ |
| **Soil S (%)** | $1.4\pm(0.7)^{A}$ | $1.4\pm(0.9)^{A}$ | $1.4\pm(0.7)^{A}$ | $1.3\pm(0.7)^{A}$ |
| **DOC (mM)** | $22.2\pm(42)^{A}$ | $1.6\pm(1)^{B}$ | $12.3\pm(14)^{AB}$ | $7.9\pm(10)^{B}$ |
| **Redox (mV)** | $193\pm(60)^{B}$ | $453\pm(58)^{A}$ | $-42\pm(98)^{D}$ | $83\pm(111)^{C}$ |
| **pH** | $7.89\pm(0.4)^{B}$ | $7.45\pm(0.2)^{C}$ | $7.96\pm(0.6)^{B}$ | $8.94\pm(0.5)^{A}$ |
| **$Fe^{2+}$ (mM)** | $0.1\pm(0.2)^{BC}$ | $0.03\pm(0.1)^{C}$ | $0.2\pm(0.2)^{AB}$ | $0.2\pm(0.2)^{A}$ |
| **Sulfide (mM)** | $0.2\pm(0.4)^{AB}$ | $0.04\pm(0.04)^{B}$ | $0.2\pm(0.4)^{AB}$ | $0.3\pm(0.6)^{A}$ |
| **Salinity (ppt)** | $12.9\pm(2.4)^{A}$ | $9.0\pm(1.8)^{BC}$ | $8.0\pm(2.1)^{C}$ | $9.6\pm(2.4)^{B}$ |
| **Total Fe (mM)** | $0.1\pm(0.1)^{B}$ | $0.1\pm(0.2)^{B}$ | $0.3\pm(0.2)^{A}$ | $0.3\pm(0.2)^{A}$ |
| **Total Ca (mM)** | $5.8\pm(1.0)^{A}$ | $5.5\pm(0.7)^{A}$ | $4.5\pm(0.9)^{B}$ | $5.3\pm(1.6)^{A}$ |


DOC concentration also varied among subsites (Table 1) and phenology (Table 2). The average

DOC concentration at SS was approximately half of that found at TS and TC, but these results are not
statistically significant due to large variability and ranges in concentration observed across the 1-year
experiment. This large variability is exemplified by standard deviations that are larger than the means. In
addition, DOC also varied across phenophases. Dormancy had the lowest mean DOC concentration and
was significantly lower than senescence by an order of magnitude. Maturity and green-up did not have
statistically different DOC concentrations. The EEMs/ UV-VIS dataset from plant maturity was analyzed
based on subsites (Table 3). There were significant differences in peaks and indices between subsites.
Coble peaks T, A, M, C and Abs$_{254}$ were significantly lower at TS than at both TC and SS by at least a
factor of two which is in line with the lower DOC concentrations observed for TS at maturity (Fig. 4).
Subsite SS had a significantly lower HIX and $E_2$:$E_3$ than both TS and TC suggesting it to have DOM with





less relative humic content and higher average molecule weight. These results indicate significantly
different DOC molecular characteristics across subsites. EEMs/ UV-VIS data could not be assessed
across phenology since these data were collected only during plant maturity.
**Table 3.** One-way ANOVA results for UV-VIS EEMs during the plant maturity phenophase. Mean
values represent average values for each subsite for subsamples from all depths. The mean is reported (±
SD) along with a connecting letter report. Means with letters that do not connect are significantly
(p<0.05) different.

| Parameter | Tall Spartina (TS) | Tall Cordgrass (TC) | Short Spartina (SS) |
|---|---|---|---|
| $Abs_{254}$ | $0.7\pm(0.2)^B$ | $1.7\pm(0.9)^A$ | $1.7\pm(1.3)^A$ |
| $SUVA_{254}$ | $0.2\pm(0.1)^A$ | $0.2\pm(0.1)^A$ | $0.2\pm(0.1)^A$ |
| $s_r$ | $1.39\pm(0.95)^A$ | $1.27\pm(0.33)^A$ | $1.46\pm(0.28)^A$ |
| $E_2:E_3$ | $5.5(0.4)^A$ | $5.4\pm(1.1)^A$ | $4.7\pm(0.7)^B$ |
| Coble T | $4.1\pm(3.8)^B$ | $14.7(10.3)^A$ | $22.6\pm(16.2)^A$ |
| Coble A | $6.6\pm(2.1)^B$ | $16.9\pm(7.02)^A$ | $13.5\pm(4.2)^A$ |
| Coble M | $4.0\pm(1.4)^B$ | $10.2\pm(4.4)^A$ | $8.6\pm(3.1)^A$ |
| Coble C | $3.7\pm(1.2)^B$ | $9.2\pm(4.0)^A$ | $7.8\pm(2.3)^A$ |
| FI | $1.3\pm(0.6)^A$ | $1.3\pm(0.02)^A$ | $1.3\pm(0.03)^A$ |
| HIX | $5.1\pm(3.0)^A$ | $4.4\pm(3.1)^A$ | $1.9\pm(0.6)^B$ |
| BIX | $0.7\pm(0.7)^A$ | $0.7\pm(0.03)^A$ | $0.7\pm(0.02)^A$ |


Differences in porewater chemistry among subsites (Table 1) and phenophase (Table 2) were also
significant. SS had the lowest average redox potential and was significantly different from TC which had
the highest, while TS was not significantly different from either SS or TC. Redox potentials were even
more variable between phenophase where all four phases had significantly different means. The highest
mean was measured during dormancy and decreased significantly in the order senescence, maturity and
green-up. The pH was not significantly different across any of the subsites but did change significantly





with phenology. Dormancy had the lowest pH which was significantly different from all other
phenophases. Senescence and green-up had a statistically similar mean pH values that were higher than
dormancy, and the porewater pH during maturity was statistically higher than all other phenophases.

$S^{2-}$ also varied significantly among subsites. SS contained on average more than an order of

magnitude greater $S^{2-}$ than both TS and TC. $S^{2-}$ is lowest during dormancy but is only significantly
different than maturity which has the highest $S^{2-}$ mean. Variability in $Fe^{2+}$ between subsites was opposite
of $S^{2-}$. While TS and TC had low concentrations of $S^{2-}$, they had high concentrations of $Fe^{2+}$, which were
more than double and significantly higher than $Fe^{2+}$ at SS. $Fe^{2+}$ concentrations varied with phenology
similar to $S^{2-}$ where dormancy had the lowest mean which was significantly different only from maturity
when the highest levels of $Fe^{2+}$ were detected. Differences between subsite total Fe followed the same
trend as $Fe^{2+}$, where SS was significantly lower than both TS and TC. Total Fe was lowest during
dormancy and senescence, which were both statistically similar, but different from green-up and maturity.

SS had the highest mean salinity and was significantly different only from TS which had the

lowest mean salinity. Green-up had a significantly lower mean salinity than all other phenophases except
dormancy. Dormancy was only significantly different from senescence, which had the highest mean
salinity. Subsite differences in Ca were similar to salinity where SS had a significantly higher mean Ca
concentration than TS, but not TC. Green-up had the lowest mean Ca concentration which was
significantly different from all other phenophases.
**3.4 Stepwise Regression Model Results**

A stepwise regression model was run across the entire 1-year experiment to determine the most

important biogeochemical predictors of soil C concentration in our dataset (Table 4). The model results
indicate that depth, redox potential, soil S, and sulfide are the best predictors of soil C concentration. The
model $R^2$ value of 0.44 indicates that these variables explain 44% of the variability in our soil C
concentration data and the model is highly significant (p <0.0001). Sulfide, redox, and soil S each have
positive estimates, meaning that these variables increase as soil C increases while depth had a negative



estimate, meaning that soil C tends to decrease with depth across the entire dataset. Each individual
predictor variable is also significant (p<0.05).

**Table 4.** Stepwise regression results for predicting soil carbon.

| Parameter | Estimate | P-Value | Model R$^2$ | Model P-Value |
|---|---|---|---|---|
| **Depth** | -0.03 | 0.003 | 0.44 | <0.0001 |
| **Sulfide** | 0.96 | 0.04 | | |
| **Redox** | 0.002 | 0.002 | | |
| **Soil S%** | 1.3 | <0.0001 | | |


**4.0 Discussion**
**4.1 Subsite Differences in Soil C and Biogeochemistry**

We hypothesized that soil C concentration and soil biogeochemistry would differ across our

subsite locations. Our results support this hypothesis and suggest significant differences in both soil C
concentration and porewater biogeochemistry among subsites, which is consistent with prior work at this
field site (Seyfferth et al. 2020; Guimond et al. 2020a). This finding illustrates the importance of
considering multiple sampling locations when conducting blue C assessments to account for ecosystem-
scale variability. At SS, average soil C concentrations were 63% higher than at TC and 29% higher than
at TS. Even though these subsites are several to tens of meters from one another, they each had
statistically different mean soil C concentrations. Higher soil C at SS is not related to higher primary
productivity because the *Spartina alterniflora* at SS are stunted. The short form of *S. alterinflora* is
generally less productive than the tall form (Roman and Daiber 1984) and likely exudes less DOC from
the smaller root mass. This is supported by a lower average DOC concentration at SS. Also, the
chromophoric dissolved organic matter (CDOM) properties at SS were different than at the other subsites.
SS CDOM had a significantly lower $E_2$:$E_2$ than TS and TC, indicative of higher molecular weight DOC at



SS. In addition, the humification index (HIX) was significantly lower at SS indicating that the DOC at SS
has been reworked by microbes less than it has been at TS and TC. Furthermore, SS consistently had
lower porewater redox potentials than the other subsites; while our data represent a snapshot in time for
each phenophase and subsite location, they are consistent with prior work of higher resolution porewater
over time that shows SS being more strongly reducing than areas closer to the tidal channel (Guimond et
al., 2020a; Seyfferth et al. 2020). Redox potentials at SS were low enough to support sulfate reduction.
This is confirmed by our elevated $S^{2-}$ porewater concentrations measured at SS. Therefore, the greatest
controls on soil C concentration at SS is slower microbial oxidation of C due to strongly reducing
conditions caused by nearly constant inundation and limited flushing of oxygenated surface water
(Guimond et al. 2020b, a; Seyfferth et al. 2020). These conditions lead to CDOM that is less affected by
microbial degradation (i.e., low HIX, low $E_2:E_3$) and a less energetically favorable metabolism (i.e.,
sulfate reduction) resulting in more C storage. This has important implications for soil C stock uncertainty
because a greater amount of the area at St Jones is composed of subsite SS (Seyfferth et al. 2020).
Sampling only near the tidal creek (TS and TC) could significantly underestimate soil C stocks, while
sampling only in the marsh interior could lead to an oversimplification of soil biogeochemistry and DOC
molecular properties in salt marsh ecosystems.

In contrast to SS, soil redox potentials were significantly higher at TC and soil C was

significantly lower. This is likely due to TC having a slightly higher elevation on a natural levee and less
reducing surface soils (Seyfferth et al. 2020). The redox potential is not low enough to support sulfate
reduction but is low enough to support Fe reduction. This is supported by the abundant amount of $Fe^{2+}$
measured in the porewater at TC. A higher redox potential and more energetically favorable electron
acceptor ($Fe^{3+}$) likely leads to higher rates of C mineralization and explains the lower soil C concentration
at TC. On the other hand, we found some of the highest concentrations of DOC at TC, particularly closer
to the surface near the rooting zone. This can be explained by a greater root mass and correspondingly
higher root exudation rate of the taller *S. cynosuroides* coupled with porewater flushing occurring only on
a spring-neap pattern, which allows DOC to build up in porewater over time (Guimond et al. 2020a, b). A



higher concentration of freshly produced DOC and a lower concentration of soil C is also consistent with
the priming effect which posits that high concentrations of freshly produced and microbially labile DOC
can stimulate microbial growth leading to the degradation of older, more stable soil C (Textor et al. 2019;
Zhang et al. 2021). In addition, TC CDOM fluorescence peaks (Coble, A, M, C, T), were similar to SS,
indicating that SS and TC have strong sources of fluorescent CDOM.

Though TS and TC are biogeochemically more similar than SS, TS had significantly higher soil C

than TC likely due to different dominant vegetation and hydrology. TS is lower in elevation and
experiences diurnal tidal oscillations with slightly lower average porewater redox values than TC (Table
1), which experiences tidal oscillations on a spring-neap cycle (Guimond et al. 2020a). These differences
in hydrology may cause soil C to accumulate more so under slightly stronger reducing conditions at TS
compared to TC. Another unique attribute of subsite TS is the CDOM signature. The coble peaks (A, T,
C, and M) and $Abs_{254}$ were significantly lower at TS than both TC and SS, which indicates a decreased
concentration of terrestrially-derived CDOM. This is likely because TS is nearest the tidal creek and
therefore porewater solutes are exported to the tidal channel twice daily during ebb tide (Fettrow et al.,
2023b), decreasing the marsh grass derived terrestrial CDOM signature in the near-channel porewater.
**4.2 Phenophase Differences in Soil C and Biogeochemistry**

We further hypothesized that soil C concentration and biogeochemistry would vary across plant

phenophase, and our data support this hypothesis. Soil C was greatest during plant dormancy and was on
average 26% higher than green-up, 18% higher than senescence, and 10% higher than maturity. This
highlights the importance of considering the time of year soil samples are taken when conducting a blue C
assessment. Likewise, many of the biogeochemical variables also changed with phenophase. The redox
potential of all four phenophases were significantly different from one another, with the highest average
redox potential occurring during dormancy. Higher redox potentials during dormancy are associated with
significantly lower porewater $Fe^{2+}$ and $S^{2-}$, indicating that microbial reduction is likely suppressed during
the winter months when labile DOC produced from root exudation is less available. Dormancy also had
the highest soil C concentration. We suggest this may be related to a suppressed priming affect due to low





porewater DOC concentrations and to Fe oxide formation during the high redox potential of dormancy,
allowing any remaining porewater C to be pulled out of solution and into the solid phase with oxidized Fe
minerals (Riedel et al. 2013; Sodano et al. 2017; ThomasArrigo et al. 2019).

We found that DOC concentrations are higher during senescence and significantly lower during

plant maturity. High porewater DOC during senescence agrees with previous work showing higher
belowground allocation of biomass in *Spartina* before the winter (Crosby et al. 2015). Belowground
allocation of C in *S. alterniflora* has been shown to increase late into the growing season (Lytle and Hull
1980) while concentrations of soil organics have been shown to decrease during the summer months due
to higher temperatures and higher rates of soil respiration (Caçador et al. 2004). Higher rates of
belowground C allocation during senescence are further supported by the higher rates of soil respiration
during senescence (Vázquez-Lule and Vargas 2021) due to increased labile DOC availability and
associated microbial activity previously reported at this field site.
**4.3 Biogeochemical Controls on Soil C**

Our data reveal important biogeochemical controls on soil C concentration across space and time.

The results of the stepwise regression model suggest that soil C concentrations are predicted by sulfide,
soil S, redox potential, and depth. Soil C increased significantly with increasing sulfide and soil S
concentration, indicated by the positive model estimate (Table 4). This is likely associated with the lower
elevation, and redox potential and greater accumulation of sulphate at SS due to less tidal flushing. This
may also be a result of sulfurization where inorganic sulfur, namely sulfide, may interact with organic
matter via abiotic reactions (Alperin et al. 1994). Evidence suggests that this interaction can help preserve
and stabilize soil C (Tegelaar et al. 1989), though spectroscopic evidence would be required to determine
if this is an important process at this study site.

Depth also has an important control on soil C concentration and the estimate was negative,

indicating that soil C decreases with depth. This is consistent with the literature suggesting higher soil C
concentration at the surface and decreasing with depth in coastal salt marshes (Bai et al. 2016). While
depth was an important predictor of soil C from the stepwise regression model, our depth profiles (Figure





4) indicate only small changes with depth. This may be a result of only sampling to 48 cm and integrating
across 12 cm increments, or it may be a result of our method design of extracting porewater from the soils
and running porewater DOC as a separate fraction of C from the solid phase soil C. Because our
porewater DOC results indicate higher concentrations near the surface, the removal of porewater DOC
prior to soil C analysis may lead to lower concentrations of soil C at the surface because in most studies,
porewater DOC is typically incorporated into the bulk soil C measurements upon soil drying and not
extracted as a separate fraction of C (i.e., porewater DOC). We suggest future studies consider porewater
DOC as a separate component of the overall soil C concentration, particularly because the variability with
depth is much higher for porewater DOC than soil C and porewater DOC is presumed to be more labile
and mobile than particulate OC. Therefore, when porewater is extracted from the soil, the measured soil C
concentration may appear less variable with depth and time leading to more consistent estimates of the
more stable solid-phase soil C.

Redox potential was the final significant predictor in the stepwise regression model and increased

significantly with soil C. We expected to see a negative relationship between soil C and redox potential
due to higher C preservation under reducing conditions, but an overall positive relationship between
redox potential and soil C in the model indicates an additional and possibly more important mechanism
related to shifting biogeochemistry throughout the year. We observed more oxic conditions at all subsites
during plant dormancy in the winter, probably due to the cold winter conditions that allow for the higher
dissolved oxygen concentrations in water and porewaters observed previously (Trifunovic et al. 2020).
Despite more oxygenated conditions and higher redox potentials in winter, the microbial activity likely
decreased during winter, allowing elevated soil C during the winter months when plants were dormant. In
addition, the less reducing and more oxygenated conditions in winter likely promoted the formation of Fe
oxides that incorporated solution-phase C into the solid phase via coprecipitation. While there is an
abundance of evidence showing the importance of Fe oxides in soil C storage in non-wetland ecosytstems
(Lalonde et al. 2012; Riedel et al. 2013; Sowers et al. 2018a, b, 2019; Adhikari et al. 2019), recent studies
have shown the important role of Fe oxides in C cycling in tidal salt marshes (Seyfferth et al. 2020;





Fettrow et al. 2023a), but few studies track C cycling during the cool winter months. Variations in Fe
oxide complexation with C due to phenological phase should be further investigated.

**4.4 Variability in Soil Carbon Storage Rates and Stocks**

Based on soil accretion rates obtained from a previous study near our core locations (Tucker

2016), bulk density at each of the three subsites previously obtained (Wilson and Smith 2015), and our
mean soil C concentrations averaged across depth for each subsite within phenophases, we calculated the
soil C accumulation rates and soil C stocks at each of the three subsites within each of the four
phenophases (Figure 7). These accumulation rates are in range of previously reported values for
mesohaline tidal salt marshes (Chmura et al. 2003; Lovelock et al. 2014; Ye et al. 2015; Mcleod et al.
2016; Macreadie et al. 2017, 2020), as are the soil C stock estimates (Zhao et al. 2016; Ewers Lewis et al.
2018; van Ardenne et al. 2018; Ouyang and Lee 2020; Gorham et al. 2021). These results further illustrate
that soil C storage rates and soil C stocks are highly dynamic and change based on time and space within
a single ecosystem. The largest difference between rates and stocks occurred between SS dormancy and
TC green-up, in which the average storage rates varied by 75% and the average stocks varied by 96%.
Therefore, within the same ecosystem and between phenophases, soil C storage rates and stocks can vary
substantially, leading to variability and uncertainty. To account for spatial and temporal heterogeneity in
soil C storage rates and stocks, we suggest taking soil cores across multiple vegetation zones (if they
exist) and across both the growing and non-growing seasons. This way, more variability can be accounted
for, leading to less uncertainty in blue C estimates.



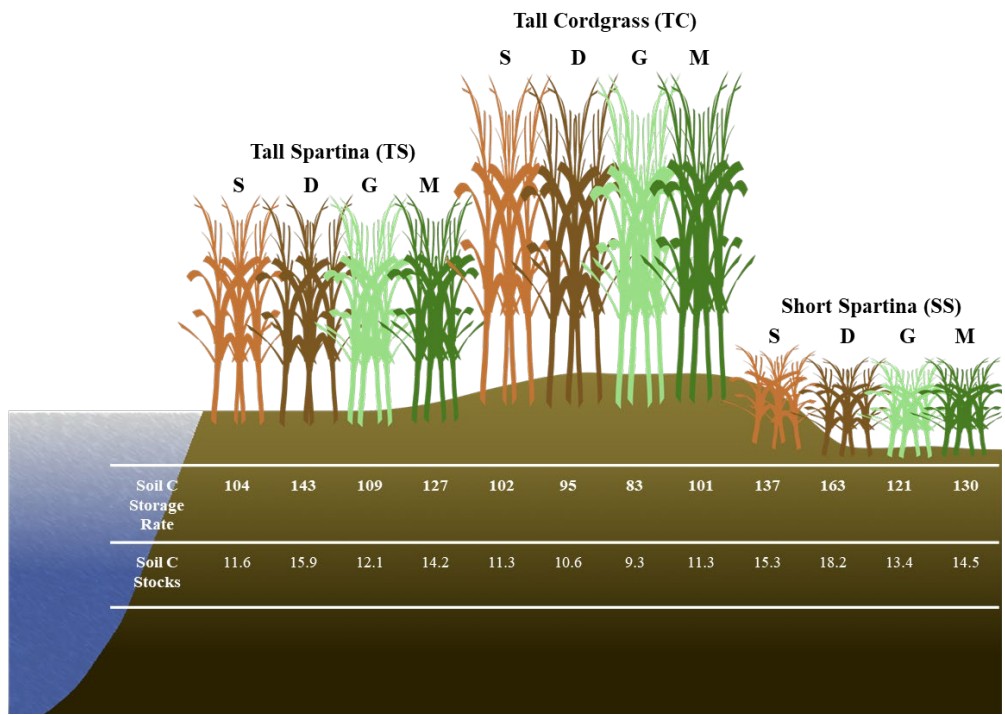


**Figure 7.** Conceptual diagram illustrating the spatial and temporal variability of soil C storage rates (g C m$^{-2}$ yr$^{-1}$) and soil C stocks (kg C m$^{-2}$) based on subsites by phenophase. Soil C stocks are to 48cm depth. S= senescence, D= dormancy, G= green-up, M= maturity.


**5.0 Conclusion**

Our results highlight the variability in soil C in time and space at the site level. We found that
some level of uncertainty in estimates of stocks and accumulation rates is likely related to spatial and
temporal variability of soil C and biogeochemistry at the marsh scale. Subsites that were only a few
meters from one another contained significantly different soil C concentrations, likely used different
metabolic pathways for C mineralization, contained significantly different porewater CDOM molecular
properties and led to considerable variation in soil C storage rates and soil C stock estimates. The
biogeochemical controls that were best correlated with soil C concentration were redox potential, soil S,
sulfide, and depth, indicating that the redox potential and sulfur content of the soils are critical in



controlling how much soil C accumulates in coastal marsh ecosystems. We also found that soil C
concentration and thus soil C storage rates and soil C stock estimates, varies significantly across the
phenophases of the marsh grasses. Plant dormancy contained the highest mean soil C concentration,
possibly a result of high redox potential during winter months that causes remaining porewater DOC to be
incorporated into the solid phase with oxidized minerals such as Fe oxides and lower microbial activity.
These results demonstrate the importance of considering marsh-scale spatial and temporal heterogeneity
when conducting a blue C assessment. Based on these results, we suggest taking soil cores from multiple
locations within a marsh and in replicate, particularly if multiple types of marsh grass are present, and at
different seasons to account for both spatial and temporal variability. These recommendations may help
lead to less uncertainty in blue C estimates.

**Statements and Declarations**
**Competing Interests:** The authors have no relevant financial or non-financial interests to disclose.
**Author Contributions:** All authors contributed to the study conception and design. Material preparation,
data collection, and analysis were performed by Sean Fettrow. The first draft of the manuscript was
written by Sean Fettrow and Angelia Seyfferth with edits by Holly Michael and Andrew Wozniak. All
authors commented on previous versions of the manuscript. All authors read and approved the final
manuscript
**Data Availability Statement**
Data is available on Figshare (DOI: 10.6084/m9.figshare.24274417)
**Acknowledgments**
We thank Chloe Kroll for laboratory assistance, the UD Soil Testing Laboratory and the Advanced
Materials Characterization Lab (AMCL) for analytical assistance, and the staff of the Delaware National
Estuarine Research Reserve (DNERR). A.L.S. and H.A.M. acknowledge support from the National
Science Foundation (Grant Nos. 1759879 and 2012484), S.F. acknowledges support from the Delaware
Environmental Institute. The authors acknowledge the land on which they conducted this study is the



traditional home of the Lenni-Lenape tribal nation (Delaware nation). The authors report no conflict of
interest.

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
