# Peer review of "Factors controlling spatiotemporal variability of soil carbon accumulation and stock estimates in a tidal salt marsh"

_EGUsphere, 2023_

## Referee Comment (RC1)

Fettrow et al. assessed C concentrations in relation to different important biogeochemical parameters at three hydrodynamically distinct sites in a coastal marsh system over the course of the course of four vegetation periods. They find that total C and DOC concentrations as well as DOC across the sites vary with biogeochemical regime across the hydrological gradient and with vegetation period. They argue that the variability in C concentrations across the sites and with depth should be taken into account when blue carbon assessments are considered.

Main comment:

The authors provide a really nice biogeochemical dataset that builds on prior in-depth characterization of the same sites ((Seyfferth et al. 2020; Guimond et al. 2020). Particularly the comparison of tidal influence versus vegetation dynamics at the three different sites yields some interesting results. I think those results are novel and useful, and should be the core of the manuscript. While I agree that C stock assessments in these kinds of systems are tricky, the study was not set up to rigorously evaluate spatial variability across these kinds of systems. Three distinct sites seem to be deliberately chosen based on their distinct hydrological regimes (see prior work cited above) and three replicate cores at each site are just not enough to truly assess spatial heterogeneity across the sites. My suggestion is to focus the revised and streamlined version of the manuscript on how tidal versus vegetation dynamics might affect total C/DOC concentration and DOC composition. There are some interesting results there that should make for a much tighter and interesting paper. In other words, focus on the spatiotemporal variability of C/DOC from a biogeochemical perspective. The fact that it makes for difficult C stock assessment is an interesting discussion point or implication, but perhaps shouldn't be the focus of this research article.

Other general comments:

- I noted below that ANOVA results are missing from Fig. 1-4. But then they were presented in Table 1-2. I suggest really shortening the results sections around Figs. 1-4 and incorporating the ANOVA results there. It is otherwise redundant, and the information provided doesn't always seem directly relevant for the questions asked and hypotheses posed.
- The authors discuss C storage rates, but what you are measuring is C concentrations and what you are estimating seems to be stocks. The term carbon storage invokes that whatever carbon is there is persistent and stored. I would only use it where appropriate. Generally, the whole text would benefit from more clearly delineating when C accrual, storage, concentrations or stocks are discussed. Or when the authors talk about pools (stocks, concentrations,…) and rates.

**ABSTRACT**

Put more emphasis on results and less on hypotheses and approach.

L41: maybe "plant phenology" instead of "ecological function"?

Maybe end on recommendation for sampling if one is interested in estimating/assessing C stocks?

INTRO

L86: exudates are by definition soluble, so perhaps omit "DOC"

Perhaps try to more clearly delineate the edaphic versus the plant controls on soil C stocks.

**METHODS**

157f: I assume the cores are extremely wet and take a long time to dry, especially given the high organic matter content. Wouldn't there by anaerobic metabolism in the glove bag, it's warm and wet in there, particularly in the presence of $H_2$?   So couldn't stem some of the seasonal variability in C content stem from differences in microbial activity at the time of sampling that then dictates how much C metabolism occurs in the glove bag?   It seems like freeze-drying might be a better alternative.

180f: Is this a water extraction or really an extraction of the residual pore water in the cores?  If it is the former, perhaps call it water extractable C. If it is the latter, isn't the extracted DOC concentration highly dependent on the moisture content at the time of sampling?  And that moisture content will be a function of where in the tidal cycles it was sampled?   Is it possible that the variability has more to do with that than site or season specific characteristics.

**RESULTS**

I don't quite understand why Fig. 2 and Fig 3 are necessary.  I think the variability is nicely illustrated by Fig. 3.  I would also add symbols indicating significant (where appropriate) in the latter.

212-214: if such a statement is made, it should be supported with adequate statistics

225-229: same as above

229-232: The regression approach is a very forgiving way to assess significant changes with depth.  I think it would be more appropriate to run a ANOVA.  But, frankly, I don't really see how they are significantly different given the large variation among the three reps.

239-243: Again, it's ok to point out trends, but if it is claimed they are different, there should be statistical tests to support that claim.

Fig. 4: I would suggest plotting DOC concentrations analogous to Fig. 3, i.e., as a box plot and run the appropriate statistics. This data is really neat and I would like to see it highlighted like that.

276-277: why isn't this discussed? Wouldn't it make sense to highlight differences across the sites as well?

Fig. 6: I don't love this figure. Could you make the lines a bit thinner so it's easier to see the individual traces? Everything is also very compressed. For example, Eh varies quite a bit with season, but it's hard to see because the scale is so compressed.

Table 1-3 header: Soil C % is not really a porewater biogeochemical variable. The table includes the solid phase.

Also, wouldn't a two way ANOVA be more appropriate to assess the influence of both vegetation and season?

378f: it would help to better explain the step-wise linear regression approach. Which factors were included and which were eliminated in the process?

---

## Author Response (AR1)

**Review #1**

Fettrow et al. assessed C concentra1ons in rela1on to different important biogeochemical parameters at three hydrodynamically dis1nct sites in a coastal marsh system over the course of the course of four vegeta1on periods. They find that total C and DOC concentra1ons as well as DOC across the sites vary with biogeochemical regime across the hydrological gradient and with vegeta1on period. They argue that the variability in C concentra1ons across the sites and with depth should be taken into account when blue carbon assessments are considered.

Main comment:

The authors provide a really nice biogeochemical dataset that builds on prior in-depth characteriza1on of the same sites ((Seyfferth et al. 2020; Guimond et al. 2020). Par1cularly the comparison of 1dal influence versus vegeta1on dynamics at the three different sites yields some interes1ng results. I think those results are novel and useful, and should be the core of the manuscript. While I agree that C stock assessments in these kinds of systems are tricky, the study was not set up to rigorously evaluate spa1al variability across these kinds of systems.

Three dis1nct sites seem to be deliberately chosen based on their dis1nct hydrological regimes (see prior work cited above) and three replicate cores at each site are just not enough to truly assess spa1al heterogeneity across the sites. My sugges1on is to focus the revised and streamlined version of the manuscript on how 1dal versus vegeta1on dynamics might affect total
C/DOC concentra1on and DOC composi1on. There are some interes1ng results there that should make for a much 1ghter and interes1ng paper. In other words, focus on the spa1otemporal variability of C/DOC from a biogeochemical perspec1ve. The fact that it makes for difficult C stock assessment is an interes1ng discussion point or implica1on, but perhaps shouldn't be the focus of this research ar1cle.

We thank the reviewer for their time in reviewing the manuscript and for the helpful comments. As the reviewer mentioned, we chose subsites based on previous research that had identified these three locations as hydrologically unique; therefore, this work extends our previous research to further investigate biogeochemical differences between the hydrologically unique subsites. We agree that we should "focus on the spatiotemporal variability of C/DOC from a biogeochemical perspective", which is what we aimed to do. In the revised version, we can revise the text accordingly to make it clear that this was our intention. We also agree that we were not able to fully spatially resolve all the spatial heterogeneity that exists in the marsh ecosystem. Doing so would require many more soil cores than we were allowed by the permit in this protected Natural Preserve area. Thus, we were only able to choose the three distinct hydrologic and biogeochemical zones to illustrate the spatial variability with replicate cores. In the revised version, we will include additional text to illustrate the limitations of the study.

**Abstract:** We have changed some of the wording in the abstract to reflect the author comment to "focus on spatiotemporal variability of C/ DOC from a biogeochemical perspective".

**Lines 124-128:** Changed some of the wording to further reflect the above author comment about spatiotemporal variability of soil C and DOC.

**Lines 163-164:** We acknowledge the limitation in this study that we cannot fully spatially resolve the variability in this ecosystem due to a limited number of samples allowed by our soil sampling permit. "We acknowledge that we cannot fully spatially resolve the variability of this

ecosystem given the limited amount of soil cores taken due to a strict soil coring permit which allowed for a specific number of soil samples".

Other general comments:

- I noted below that ANOVA results are missing from Fig. 1-4. But then they were presented in Table 1-2. I suggest really shortening the results sec1ons around Figs. 1-4 and incorpora1ng the

ANOVA results there. It is otherwise redundant, and the informa1on provided doesn't always seem directly relevant for the ques1ons asked and hypotheses posed.

We thank the reviewer for their feedback and we can work to shorten the results section to remove any redundancy. It should be noted that Fig 1 is a map of soil core locations; therefore, there is no ANOVA results for the map. Also it should be noted that the information in Figures 2, 3, and 4 are not the same thing as those reported in the ANOVA tables in Tables 1 and 2.

Figure 2 reflects depth profiles of individual replicate soil C measurements at each depth at each phenophase and each subsite so that readers can visually understand the specific variability of
soil C at each depth, phenophase, subsite and replicate core, all in one figure. In contrast, Table 1 shows the ANOVA results of total C in the entire depth profiles and phenophase but separated by subsite/ vegetation zonation as well as 9 other biogeochemical variables that are not represented in Figure 2.

The goal of Figure 3 is to show the reader ranges in soil C and S concentration by depth but separated by subsite, while Table 2 shows the ANOVA results of total C and S as well as 8 other biogeochemical variables by subsite where depth and phenophase are averaged. Similarly, Figure
4 shows variability of DOC as a heatmap across all replicate cores and phenophases, depths and subsites. As the author points out, this paper is about the variability of soil C and DOC so we want to show how variable it is at different scales by showing all replicate points presented in a visual, easy to identify way, just as we have done with soil C.

We have still gone through all the results and trimmed out any redundancies where necessary and also incorporated the authors first comment about "spatiotemporal variability of soil C in the context of biogeochemistry". We have also included a new 3-way ANOVA analysis that helps to better summarize the porewater chemistry and solid phase C and S findings.

- The authors discuss C storage rates, but what you are measuring is C concentra1ons and what you are es1ma1ng seems to be stocks.

Yes, we use our C concentration values to calculate stocks, because we know the bulk density of these subsite locations from previous research, which was cited in the paper (Wilson and Smith
2015) and stated in the paper in lines 512-516. In the revised version, we will be sure to state them as such. We also use previously estimated soil accretion rates to calculate soil C storage rates (Line 512-513).

We clarified in the revised manuscript that we measured soil C and estimated accrual rates based on previous sedimentation data (Wilson and Smith 2015) and we estimated stocks based on previous bulk density data (Tucker 2016). We simplify these throughout the text to only say "Soil C accrual" when discussing rate and only say "Soil C stock" when discussing stock.

The term carbon storage invokes that whatever carbon is there is persistent and stored. I would only use it where appropriate. Generally, the whole text would benefit from more clearly delinea1ng when C accrual, storage, concentra1ons or stocks are discussed. Or when the authors talk about pools (stocks, concentra1ons,…) and rates.

We agree that terminology around the subject of soil C storage/ stocks can be confusing and

often lacks a clear definition in each context. We agree that this terminology should be simplified
in the text. In a revised version, we will only using the term "soil C accrual rate" when discussing (g C m$^{-2}$ yr$^{-1}$) and only using the term "soil C stocks" when discussing (kg C m$^{-2}$). Generally, this was our intention, but we agree there are a few areas where terminology can be better clarified.

A word search was done for the entire manuscript and "storage" was changed to "accrual" throughout the text when discussing "Soil C accrual" rates.

ABSTRACT

Put more emphasis on results and less on hypotheses and

approach. We agree and will be sure to do this in the revised

paper.

We have revised the abstract to remove some of the hypothesis/
methods wording and focus more on the main results/ findings.

L41: maybe "plant phenology" instead of "ecological func1on"?

Yes, we agree that "ecological function" should be replaced with plant phenology on Line 41.

Line 41: "Ecological function" has been replaced by "plant phenology".

Maybe end on recommenda1on for sampling if one is interested in es1ma1ng/assessing C stocks?

We agree we should end the abstract on a recommendation, and we feel that is already at the end of the abstract. "It is, therefore, critical to consider spatial and temporal heterogeneity in soil C concentration when conducting blue C assessments to account for soil carbon variability and uncertainty in C stock estimates". In a revised version we could also add further detail such as "we recommend that multiple locations and timepoints are sampled when conducting blue C studies to account for ecosystem-scale variability".

Line 43: We have added this recommendation to this end of the abstract.

INTRO

L86: exudates are by defini1on soluble, so perhaps omit "DOC"

We understand how this line might be confusing, but we want to introduce both "root exudates" and "DOC" into the story, and root exudates are just one source of DOC. We suggest rewording these sentences to "Belowground production of dissolved organic carbon (DOC) can come from root exudation (Luo et al. 2018)".

Line 84: We have changed the wording of this phrase to make is flow better and is easier to understand, as the reviewer pointed out.

Perhaps try to more clearly delineate the edaphic versus the plant controls on soil C stocks.

We thank the reviewer for this comment and agree we should add/ edit the intro to distinguish between plant vs. soil related effects more clearly on soil C stocks. In a revised version of the manuscript we will make those additions.

We distinguished between environmental controls, soil controls, and plant controls more clearly in the introduction section.

**METHODS**

157f: I assume the cores are extremely wet and take a long 1me to dry, especially given the high organic ma#er content. Wouldn't there by anaerobic metabolism in the glove bag, it's warm and wet in there, par1cularly in the presence of H2?

After taking the field-moist samples for total C and S analysis, the soils were capped in their vials to remove the porewater with centrifugation. Afterward, the remaining soil was placed in the glove bag using large trays of desiccant to rapidly remove residual moisture from the drying soils. The glove bag itself would not be "wet" because the desiccant is constantly removing moisture from the air and we monitor the % humidity that is often <<20%. We regenerate the desiccant daily to quickly remove water from the soils while also slowing down fast oxidative reactions.

So couldn't stem some of the seasonal variability in C content stem from differences in microbial ac1vity at the 1me of sampling that then dictates how much C metabolism occurs in the glove bag? It seems like freeze-drying might be a be#er alterna1ve.

That is possible, but all of the samples were treated in the same way. So, if H2-fueled metabolism happened in the glovebag, it would have affected all cores equally and not be the main driver of the differences across sites or seasons. Also, the large size of the cores prevented us from being able to freeze-dry them with the equipment available. In the revised version, we can discuss this point as a possible limitation of the study.

Line 179- 180: We acknowledge that drying inside the glovebag may have created some level of H2-fuled metabolism, but likely didn't last long because we rapidly dried the soils with fresh desiccant.

180f: Is this a water extrac1on or really an extrac1on of the residual pore water in the cores? If it is the former, perhaps call it water extractable C. If it is the la#er, isn't the extracted DOC concentra1on highly dependent on the moisture content at the 1me of sampling? And that moisture content will be a func1on of where in the 1dal cycles it was sampled? Is it possible that the variability has more to do with that than site or season specific characteris1cs.

We extracted porewater by centrifuging the core, without any additives such as deionized water, so based on the reviewer's definition it would be "residual pore water" but we just refer to it as pore water.

The amount of porewater we obtain is a function of saturation and to be consistent across sampling timepoints, we cored the locations at the same tidal inundation cycle each season. We will be sure to better articulate this in the text.

Lines 179-180: We articulated the point that porewater volume is a function of saturation and to keep this consistent across sampling timepoints, we cored at the same tidal inundation cycle each season.

**RESULTS**

I don't quite understand why Fig. 2 and Fig 3 are necessary. I think the variability is nicely illustrated by Fig. 3. I would also add symbols indica1ng significant (where appropriate) in the la#er.

The purpose of Figure 2 was to show how C concentrations in individual cores varied in space and time. Figure 3 is a summary of that variability in C and S with depth at all timepoints together. We could put Figure 2 in the supporting information, but we would worry that we are then not including the variability in C between cores that were intended to be replicates. Fig 2 represents the large amount of Soil C variability on different spatial (depth, replicate, subsite) and temporal (seasons) scales.

We save ANOVA significant results for a table when we assess overall summarized differences in all variables between subsite and phenology, but we agree that Fig 3 would benefit from having a significant difference letters report. We will be sure to add in sig differences to Fig 3 in a revised version of the manuscript.

After running ANOVAs to look for differences between depths (for Fig 3 averaged across replicates/ seasons for each subsite), only soil S has significant differences between depths when averaged across depth/ subsite. We have indicated those significant differences in the soil S panel, and indicate that dynamic in the text as well (Lines 231-234).

212-214: if such a statement is made, it should be supported with adequate sta1s1cs 225-229: same as above

Note we did not say "significantly higher" in this sentence, we just note that its higher, but we can include modifier and say "appeared to be higher", which is later backed up statistically by the ANOVA results in Tables 1 and 2.

We changed "higher" to "appeared to have higher", since this figure just visually shows variability. The statistics come later in the manuscript.

229-232: The regression approach is a very forgiving way to assess significant changes with depth. I think it would be more appropriate to run a ANOVA.

We thank the reviewer for their comments on our statistical approach. We could try a 2-way ANOVA with depth and subsite as factors for each of the phenophases. We could also attempt a 3-way ANOVA with depth, phenophase, and subsites as factors, and provide the results in a revised version of the manuscript

We have revised the ANOVAs and tables. Table 1 now contains the three-way ANOVA with subsite, phenology and depth as factors, as well as the interaction between all three factors. Tables 2 and 3 contain the individual means and standard deviations with connecting letter reports for the post-hoc Tukey test.

But, frankly, I don't really see how they are significantly different given the large varia1on among the three reps.

The purpose of the Figure 2 heatmaps is to show the variation in soil C with subsite, phenophase, and depth, and contrast those with Table 1's ANOVA results that are grouped by subsite and averaged across phenophase and depth. When grouped by subsite and averaged across phenophase and depth, there are significant differences across subsites (Table 1) but when grouped by phenophase and averaged across depth subsite, there are fewer significant differences (Table 2).  The ANOVAs presented in Tables 1 and 2 clearly illustrate significant differences with the connecting letter report.

We have reorganized and reran the ANOVAs to start with the three-way ANOVA to consider all three factors relative to one another and how they affect each variable, along with their

interaction. We still include the post-hoc Tukey tests because this clearly shows the individual significant differences at the subsite, phenology and depth (supplemental) factor levels.

239-243: Again, it's ok to point out trends, but if it is claimed they are different, there should be sta1s1cal tests to support that claim.

In the revised version, we will be sure to clarify that these are trends. We can also run the ANOVA on these Figure 3 and include those as well.

Fig 3 has been modified to include sig letter report.

Fig. 4: I would suggest plokng DOC concentra1ons analogous to Fig. 3, i.e., as a box plot and run the appropriate sta1s1cs. This data is really neat and I would like to see it highlighted like that.

We agree and will include that in the revised version.

Figure 5 is now a plot similar to Fig 3.

276-277: why isn't this discussed? Wouldn't it make sense to highlight differences across the sites as well?

We saved the discussion for the discussion section and discussed differences across the subsites on lines 391-445.

Fig. 6: I don't love this figure. Could you make the lines a bit thinner so it's easier to see the individual traces?

We made the markers different shapes and colors so its easier to follow the line and keep track of sample site location with depth. We tried to make the lines thinner and plot markers slightly smaller, but when shrunk down to PDF size they were unreadable.

Everything is also very compressed. For example, Eh varies quite a bit with season, but it's hard to see because the scale is so compressed.

This figure is a compromise between trying to show all of the data for each of the variables at all times and space verses showing the overall trends by season. What we hoped the reader would get out of this figure is that something like Eh varies substantially by season even more than by subsite. The Eh scale ranges from -400 to 600 so the scale is quite broad. This is so that we could fit all the variability across seasons and subsites onto one scale, rather than making a different range in scale for each figure. This way, the reader can more easily compare the broad differences in redox across sites, seasons and depth.

Table 1-3 header: Soil C % is not really a porewater biogeochemical variable. The table includes the solid phase.

We thank the reviewer for this comment. We agree that Soil C and S is not a porewater variable, and in a revised version we can edit the table header from "porewater biogeochemical variables" to "soil and porewater chemical variables".

We edited the header of the tables reflect this comment.

Also, wouldn't a two way ANOVA be more appropriate to assess the influence of both vegeta1on and season?

We can try a two-way ANOVA to look at the combined influence and interactions of both subsite and season. We will include those results in a revised version.

As described above, based on the reviewer comment we decided to redo our ANOVAs with a three-way ANOVA approach, followed by the same post-hoc Tukey results. So table 1 is now the three-way results while tables 2 and 3 are the same post-hoc analysis.

378f: it would help to be#er explain the step-wise linear regression approach. Which factors were included and which were eliminated in the process?

We thank the reviewer for this comment and agree the step wise regression approach could use more detail. We used all variables listed in Table 1 and 2 and the stepwise regression model was run to maximize the $R^2$ while using the least amount of variables to explain the variance. That is, the model was run to determine the most important (significant) biogeochemical variables we measured for predicting soil C concentration. In a revised version of this manuscript, we will add more detail about the variables used in the regression model, and how these final variables were chosen to represent the final model.

Line 204: We state these details about how the final stepwise regression model was determined.

Main comment

The authors provide important insights into soil carbon variability based on the biogeochemical characterisation of the study site. The results are interesting and valuable for the 'blue carbon community' and emphasise the thorough investigations needed to assess carbon stocks in tidal marsh habitats accurately. Some clarifications would benefit the manuscript.

- •Most blue carbon stock and sequestration rate studies focus on Total Organic Carbon concentrations. The manuscript focuses on total soil carbon (assuming this is the term the authors use for both organic and inorganic components). Organic and inorganic carbon fractions in soil can react differently under redox conditions. It's important to make this distinction.

    We thank the reviewer for this comment. The reviewer is correct in their assumption that we report soil C as Total soil C, and that we did not separate soil C into organic vs inorganic. In a revised version of the manuscript we will clarify that we report Total Soil C, and to make the comment that inorganic and organic are two separate pools of C that react differently under variable redox conditions, but we only measure total C in this study. We will highlight this limitation in our study.

    We made this edit and clarification on line 177.

- •There seems to be some confusion regarding the terminology relating to carbon stocks vs concentrations. The methodology section covers soil C % calculations. However, the core stock calculations are missing and are only mentioned in section 4.4. If other data is used from previous publications to calculate soil carbon stocks, this should be detailed in the methodology.

    We thank the reviewer for this comment and agree there should be a clearer explanation of soil C stock calculations in the methods. We do mention that other values (bulk density) were used from a previous study (Line 513) to calculate soil C stock, but we agree this information should also be mentioned in the methods near the soil % calculations and will include it in a revised version of the manuscript.

Line 178: We explain that we calculate soil C stocks from previously obtained bulk density and soil C accrual rates from previously obtained sedimentation rates at our field site.

- •The statistical methods chosen in the manuscript need more clarification, perhaps in the supplementary materials. Were the assumptions of equal variance and normality met before proceeding with ANOVA? What criteria were used for the subset of predictors in the stepwise regression model? Why was the stepwise regression model chosen over partial least squares regression?

    Assumptions of ANOVA were met by assessing for normality with QQ plots and we transformed the data to achieve normality when necessary ($Fe^{2+}$, $S^{2-}$, DOC, and Total Fe). Equal variance was tested to ensure homogeneity in variance between subgroups with a

Levene's test. In the revised version, we will include this information as well as the criteria used for the stepwise regression model predictors.

Lines 200-203: We explain how assumptions of ANOVA were met. We also further explain stepwise regression model selection.

Also, we used stepwise regression because in our program (JMP), it cycles through every combination of factors through iterative loops to arrive at the best (i.e., best $R^2$ and least amount of explanatory variables) possible model. It is best used for model selection and removes any sort of bias towards selecting specific predictor variables.

- In the conclusion sections of the manuscript, the authors make sampling recommendations for carbon stock assessments. It would be useful to briefly compare with existing sampling guidance e.g., Howard et al., 2014 and Bansal et al., 2023. These guidance documents recommend sampling based on marsh zonations and at the time of highest plant biomass (late summer). Given that most guidance documents aim to provide sampling for quantification of long-term TOC carbon pools, perhaps it's better to emphasise the importance of sampling from multiple locations within the marsh, rather than seasonal sampling.

We thank the reviewer for this comment and agree that reviewing our recommendations alongside other recommendations would make this article more useful and meaningful. We will add a discussion to the end of section 4.4, when discussing soil carbon storage rates and stocks and further discuss our recommendations for accounting for this ecosystem scale variability rather than seasonal sampling.

Line 526: We further state our recommendations alongside Howard et al., 2014 which discusses in detail recommendations and procedures for accounting for soil C stocks and storage in vegetated coastal ecosystems.

Other comments

Please check the manuscript for consistency with abbreviations e.g., 12 cm vs 12cm vs 12-cm.

We thank the reviewer for this comment. In a revised version of the manuscript, we will ensure that we are consistent with abbreviations and units.

We have checked all unit abbreviations to make sure there is a space in between the number and the unit and that all abbreviations are consistent.